# Minimal Semantic Sufficiency
# Meets Unsupervised Domain Generalization

**Tan Pan**[1,2,*] **Kaiyu Guo**[3,2,*] **Dongli Xu**[2] **, Zhaorui Tan**[2] **, Chen Jiang**[1,2,†] **Deshu Chen**[1]
**Xin Guo**[1,2] **, Brian C. Lovell**[3] **, Limei Han**[1,2] **, Yuan Cheng**[1,2,†] **Mahsa Baktashmotlagh**[3]
[1] AI[3], Fudan University [2] Shanghai Academy of Artificial Intelligence for Science
[3] The University of Queensland
pant23@m.fudan.edu.cn, jiangchen@sais.org.cn, cheng_yuan@fudan.edu.cn

## Abstract

The generalization ability of deep learning has been extensively studied in supervised settings, yet it remains less explored in unsupervised scenarios. Recently, the Unsupervised Domain Generalization (UDG) task has been proposed to enhance the generalization of models trained with prevalent unsupervised learning techniques, such as Self-Supervised Learning (SSL). UDG confronts the challenge of distinguishing semantics from variations without category labels. Although some recent methods have employed domain labels to tackle this issue, such domain labels are often unavailable in real-world contexts. In this paper, we address these limitations by formalizing UDG as the task of learning a **M**inimal **S**ufficient **S**emantic **R**epresentation: a representation that (i) preserves all semantic information shared across augmented views (sufficiency), and (ii) maximally removes information irrelevant to semantics (minimality). We theoretically ground these objectives from the perspective of information theory, demonstrating that optimizing representations to achieve sufficiency and minimality directly reduces out-of-distribution risk. Practically, we implement this optimization through Minimal-Sufficient UDG (MS-UDG), a learnable model by integrating (a) an InfoNCE-based objective to achieve sufficiency; (b) two complementary components to promote minimality: a novel semantic-variation disentanglement loss and a reconstruction-based mechanism for capturing adequate variation. Empirically, MS-UDG sets a new state-of-the-art on popular unsupervised domain-generalization benchmarks, consistently outperforming existing SSL and UDG methods, without category or domain labels during representation learning.

## 1 Introduction

Generalization ability is a critical yet challenging problem in deep learning. This challenge leads to the emergence of the task of Domain Generalization (DG) [55, 53, 42], which focuses on generalizing deep learning models to unseen distributions. Many works have been proposed on this topic under the supervised settings [47, 25]. However, for the scenario of unsupervised learning, *e.g.,* self-supervised learning (SSL), which is more prevalent and practical in the real world [33, 13, 41], limited works have been proposed to address the generalization ability of the model. Recently, the task of Unsupervised Domain Generalization (UDG) [56] has been proposed to handle the generalization issue in self-supervised learning (SSL), and some methods have been proposed to improve the generalization ability by learning or disentangling the domain-invariant representations [54, 56, 51]

---

*Equal contribution. This research was conducted during an internship at the Shanghai Academy of Artificial Intelligence for Science.
† Corresponding authors.

However, two key challenges remain. (i) Traditional DG methods [31, 29] often focus on disentangling semantics from variation factors [55], which becomes more challenging in UDG due to the absence of category labels (*e.g.*, cat, dog, and airplane). (ii) In order to disentangle meaningful semantics from domain-specific variations, existing UDG techniques predominantly depend on domain labels (*e.g.*, painting, scratch, and clipart) that are inaccessible or expensive in real-world scenarios. Motivated by these challenges, our work addresses a critical research question: *How can semantic information be effectively disentangled in UDG without relying on domain labels?*

To address this research question, we explore learning semantics from an information-theoretical perspective [2, 48]. Specifically, we argue that the shared information learned by contrastive learning is adequate to represent semantics [44]. However, it may also encompass semantically irrelevant but confounding factors, such as shared style and texture, which can lead to suboptimal semantic representations. This issue might be exacerbated when unlabeled data exhibit significant semantic and covariate distribution shifts. Therefore, eliminating semantically irrelevant information is crucial to refining sufficient semantic representations for UDG.

Based on the above assumption, our intuition is to separate the semantic and variation information by minimizing semantically irrelevant information within sufficient representations. To achieve this, we formulate this non-trivial challenge as a constrained optimization problem, aiming to reduce the dependency between the sufficient representation and the input conditioned on the semantic space. Concretely, the optimization problem can be decomposed into two objectives: (1) minimizing the mutual information between the semantic and variation representations, and (2) maximizing the mutual information between the variations and the inputs conditioned on the semantic space. Objective 1 ensures that semantics and variations are disentangled, promoting their independence and non-redundancy. Objective 2 enables the semantic representation to retain minimal semantically irrelevant information. By optimizing these two objectives, the system is able to learn approximately optimal semantics that is both disentangled and minimally sufficient.

To achieve these objectives, we introduce a novel algorithm, namely Minimal Sufficient UDG (MS-UDG). The model first employs traditional contrastive learning to ensure a sufficient semantic representation. Then, two modules, *i.e.*, Information Disentanglement Module (IDM) and Semantic Representation Optimizing Module (SROM), are employed to disentangle the optimal semantic representation from the sufficient semantic representation. IDM separates the semantic and variation representations from the sufficient semantic representation. SROM applies a mixed InfoNCE constraint to minimize the mutual information between semantics and variations. Simultaneously, it maximizes the mutual information between variations and the inputs through reconstruction, thereby fulfilling our learning objectives.

**Contributions.** (1) To the best of our knowledge, we propose the first theoretical optimal semantic representation for UDG from an SSL perspective. Subsequently, we introduce a tractable estimation method for disentangling the optimal semantic representation through two optimization objectives. This approach offers a novel view for enhancing the generalization capabilities of SSL models to previously unseen domains.

(2) Based on dual optimization objectives, we introduce MS-UDG, an algorithm that effectively removes semantically irrelevant information while preserving representative semantics, without relying on domain labels. We also provide theoretical analysis to support the rationale and applicability of our framework.

(3) Experimental results on popular benchmark UDG datasets demonstrate that our method achieves superior performance in downstream tasks compared to existing approaches.

## 2 Related Work

**Unsupervised Domain Generalization.** Unsupervised Domain Generalization (UDG) has been proposed to handle the problem of domain generalization in unsupervised learning [56]. Similar to supervised domain generalization [30, 28, 20, 55, 15], methods in UDG mostly rely on representation learning and data augmentation to conduct domain-invariant or domain-specific self-supervised learning (SSL). For data augmentation, FDA [50] and BSS [39] introduce Fourier-based methods to standardize the style of images to plug in native contrastive-based SSL. In addition, BrAD[16] generalizes the model by aligning the image to a unified style on the feature level. For representation

learning, contrastive-based methods construct negative samples [56] or suppress intra-domain connectivity by domain labels [27]. Also, MAE [17] is utilized in UDG. DiMAE [52] and CycleMAE [51] transform the original image into its style-mixed view and then decode different domain styles by several domain decoders. DisMAE [54] introduces a semantic-encoder and a variation-encoder to disentangle semantic attributes. Moreover, some methods use UDG methods to improve the generalization on specific tasks, such as face anti-spoofing [26]. Unlike these methods, we theoretically build our method from the contrastive learning perspective.

**Contrastive Learning.** Contrastive learning [33, 5, 34] is a successful paradigm in SSL. Contrastive-based SSL methods aim to learn the shared information [32] between multi-view data. Recent methods [46, 44, 24] discuss task-relevant and redundant representations of SSL based on information theory. Although there has been extensive exploration in the SSL field, the generalization of SSL to OOD data still requires further discussion.

**Disentangled Representation Learning.** Disentangled representation learning is a promising approach for DG, aiming to separate semantics-relevant and irrelevant factors in data [49, 56, 21, 57**?**]. Class labels help ensure the model learns class-specific features consistently across domains, avoiding reliance on domain-specific artifacts. Without class labels, the disentanglement-based methods in UDG often focus on domain style transfer [53, 55] to separate domain variations by domain labels. However, there is still a lack of theoretical support for disentangling domain-invariant representations of SSL to obtain optimal semantics.

# 3 Preliminary and Motivation

In Sec. 3.1, we first introduce the setting of UDG, which aims to discard the covariate information in the representations [54]. To address our research question, in Sec. 3.2, we propose a new concept of minimal sufficient semantic representation, treating semantic information as a proxy for the unknown downstream task. Based on this concept, we theoretically formulate two learning objectives optimized to disentangle representation into pure semantic information. Finally, based on these objectives, we present our algorithm and provide a theoretical analysis in Sec. 4.

## 3.1 Unsupervised Domain Generalization

**Notation.** Let $\mathcal{X}$ be the input space and $\mathcal{Y}$ be the label space. Considering a supervised dataset $D = (X_D, Y_D)$ with $N_D$ samples, where $X_D \subset \mathcal{X}$ and $Y_D \subset \mathcal{Y}$, the distribution of $D$ is $P_{(X_D, Y_D)}$. Then we can define the distribution of SSL data, downstream supervised data, and test data as $P_{(X_{ssl}, \emptyset)}$, $P_{(X_{sup}, Y_{sup})}$ and $P_{(X_{test}, Y_{test})}$ respectively. In SSL, for each input $(x_1, x_2)$, we have $x_1 \in X_{ssl}$ and $x_2 \in X_{aug}$, where $X_{aug}$ is augmented from $X_{ssl}$. The extracted features are $z_1 = h(x_1)$ and $z_2 = h(x_2)$, where $h$ is the encoder and $z_1, z_2 \in \mathcal{Z}$, $\mathcal{Z}$ is the representation space. The parameterized latent space for semantic-relevant and -irrelevant, or variate, factors are $\mathcal{S}$ and $\mathcal{V}$, respectively. The intuitive relation is $\mathcal{Z} = \mathcal{S} \oplus \mathcal{V}$.

**Problem setting.** Following the setting of UDG [56], the test distribution should be kept unknown at any stage before inference. Then, we have the condition $Support(P(X_{test})) \cap (Support(P(X_{sup})) \cup Support(P(X_{ssl}))) = \emptyset$. In addition, to avoid the semantic shift in the downstream task, we need $Support(P(Y_{sup})) = Support(P(Y_{test}))$. The target of UDG is to obtain the optimized feature extractor $h^*$ which has the minimum risk on the test distribution $h^* = \arg\min_h R_{P_{(X_{test}, Y_{test})}}(h)$

**Motivation.** Considering that UDG methods are built upon existing SSL methods with unknown downstream task targets $T$, we examine the goal of UDG from a contrastive learning (CL) perspective. Given an input $x_1$ and its augmented counterpart $x_2$, we adhere to the multi-view assumption [44], treating $x_1$ and $x_2$ as two corresponding views of the same data point. Under this assumption, the objective of CL is to extract task-relevant representations [44]. [44] posits a compression gap $I(x_1; x_2 | T)$ under the task $T$. This implies that while optimizing for $I(x_1; x_2)$, task-irrelevant information may inadvertently be retained in the learned representations. In datasets used for SSL, especially those from multiple domains that exhibit significant semantic and covariate distribution shifts, this compression gap cannot be overlooked if the aim is to achieve task-specific or semantic-specific representations. Based on this insight, we propose that semantics and variations can be separated by explicitly modeling and minimizing task-irrelevant information.

## 3.2 Minimal Sufficient Semantic Representation

In this section, we first introduce the definition of sufficient representation and minimal sufficient representation [46, 44], which aims to learn task-relevant representation optimally. In the supervised setting, with the given image $x$ and label $y$, the information bottleneck [43] can be applied to achieve optimal representations by minimizing mutual information $I(x, z)$ and maximizing $I(z, y)$, where $z$ is the representation of $x$. In self-supervised learning, the augmented image $x_2$ plays a similar role as the supervised label [11, 44]. The definition is proposed as follows.

**Definition 3.1** (Minimal Sufficient Representation in Contrastive Learning [46]). Let $\hat{z}_1^{suf}$ and $\hat{z}_1^{min}$ be the sufficient representation and minimal sufficient representation of $x_1$ in self-supervised learning, respectively. $x_2$ is the augmented data. Then the following conditions should be satisfied.

$$I(\hat{z}_1^{suf}; x_2) = I(x_1; x_2), \hat{z}_1^{min} = \arg\min_{\hat{z}_1^{suf}} I(\hat{z}_1^{suf}; x_1)$$

This definition introduce the minimal sufficient representation with two stage: First, we define that $\hat{z}_1^{suf}$ is sufficient if and only if $I(\hat{z}_1^{suf}; x_2) = I(x_1; x_2)$, which aligns with the target of contrastive loss to maximize $I(z_1; z_2)$ [32]. Note that, since $\hat{z}_1^{suf}$ is defined to keep all the mutual information between $x_1$ and $x_2$ instead of any prior labels, the loss or change of augmented information in $x_2$ can also affect $\hat{z}_1^{suf}$. Second, with $\hat{z}_1^{suf}$, we define the minimal sufficient representation $\hat{z}_1^{min}$ by searching minimum $I(\hat{z}_1^{suf}; x_1)$. Symmetrically, we can also define the minimal sufficient representation of $x_2$. More details of minimal sufficient representation can be found in Appendix.

However, the minimal sufficient representation in Definition 3.1 may still include task-irrelevant information, particularly in UDG datasets with significant covariate shifts.

**Proposition 3.2.** $I(\hat{z}_1^{min}; x_i) = I(\hat{z}_1^{min}; T) + I(\hat{z}_1^{min}; x_i|T) \geq I(\hat{z}_1^{min}; T)$, where $i \in \{1, 2\}$, $T$ is the downstream task.

The proposition 3.2 indicates that if $I(x_i; \hat{z}_1^{min}|T)$ is large, $\hat{z}_1^{min}$ may not be the optimal minimal sufficient feature. In practice, $I(x_i; \hat{z}_1^{min}|T)$ can be affected by many factors in the shared information, including style, texture, and other factors between $x_2$ and $x_1$. In the multi-domain situation as the setting of UDG, the $I(x_i; \hat{z}_1^{min}|T)$ will be large, which leads to performance degradation since there can be significant distribution shifts in all task-irrelevant distributions.

The downstream task remains unseen during SSL. However, in most cases, it should be related to the semantic information in $X_{sup}$. In this case, we can consider $I(x_1; x_2; T) = I(x_1; x_2; S)$. Then, we argue that the sufficient representation should contain all semantic information between $x_1$ and $x_2$.

**Definition 3.3** (Sufficient Semantic Representation in Contrastive Learning). $z_1^{suf}$ is the disentangled sufficient representation of $x_1$ if and only if $I(z_1^{suf}; x_2; S) = I(x_1; x_2; S)$. $S \subset \mathcal{S}$ is the semantic information in $X_{ssl}$

Definition 3.3 indicates that a representation containing all the shared semantic information between $x_1$ and $x_2$ is sufficient to capture the semantics of $x_1$ and $x_2$. However, the sufficient representation may not be the optimal representation as $I(z^{suf}; x|S) \geq 0$, where $I(z^{suf}; x) > 0$.

From the relation $\mathcal{Z} = \mathcal{S} \oplus \mathcal{V}$, every $z^{suf}$ can be decomposed as $z^{suf} = (s, v)$, where $s$ is semantic-relevant, which is also a sufficient semantic representation, and $v$ is semantic-irrelevant. That means $I(v; S) = 0$. By this disentanglement, if $I(s; x|S)$ can be minimized, then $s$ is the optimal sufficient representation we expect. From this perspective, we propose the definition of the optimal sufficient semantic representation in Definition 3.4.

**Definition 3.4** (Minimal Sufficient Semantic Representation in Contrastive Learning). Let $z_1^{min}$ be the minimal sufficient semantic representation of $x_1$. Then, $z_1^{min} \triangleq \arg\min_{z_1^{suf}} I(z_1^{suf}; x_1, x_2|S)$, $S \subset \mathcal{S}$ is the semantic information in $X_{ssl}$, $x_2$ is the augmented input.

Definition 3.4 suggests that the minimal sufficient semantic representation should contain the least semantic-irrelevant information. Furthermore, we can assume $I(z_1^{min}; x_1, x_2|S) = 0$. As mentioned above, we have the decomposition of sufficient semantic representations $\forall z^{suf}, z^{suf} = (z^{min}, v)$. Then, we theoretically illustrate how to effectively disentangle the $z^{min}$, which answers the research question we proposed.

**Proposition 3.5.** $I(z_1^{suf}; x_1 | S) = I(z_1^{suf}; x_1, x_2 | S)$

Proposition 3.5 indicates that the semantic-irrelevant information in $z_1^{suf}$ is derived from $x_1$, and we can obtain $z_1^{min}$ as $z_1^{min} \triangleq \arg\min_{z_1^{suf}} I(z_1^{suf}; x_1 | S)$,

**Proposition 3.6.** $z = (s, v)$ *is the sufficient semantic representation of* $x$ *and* $s$ *is semantic-relevant representation. Then* $I(s; x | S) = I(s; v) + I(z; x | S) - I(v; x | S)$

If we want to optimize $s$ as the minimal sufficient semantic representation, considering the Markov chain $x \rightarrow z \rightarrow s, v$, we can only optimize $s$ and $v$. ***So, we propose to minimize*** $I(s; v)$ ***and maximize*** $I(v; x | S)$ ***to disentangle the optimal semantic representation in UDG.***

## 4 Algorithm and Theory

### 4.1 Algorithm

Based on the objectives in Proposition 3.6: (1) minimize $I(s; v)$ and (2) maximize $I(v; x | S)$, we propose that objective 1 enhances the independence and non-redundancy between semantic representation and variations, while objective 2 limits the inclusion of semantic-irrelevant information in semantic representation.

To achieve objectives, we propose the method Minimal Sufficient UDG (MS-UDG) as shown in Fig. 1. After learning sufficient representations by contrastive learning loss InfoNCE [18], the model comprises two components: (1) Information Disentanglement Module (IDM), which aims to disentangle semantic representation $s$ and variation $v$. (2) Semantic Representation Optimizing Module (SROM), which optimizes $s$ towards minimal sufficient semantic representation.

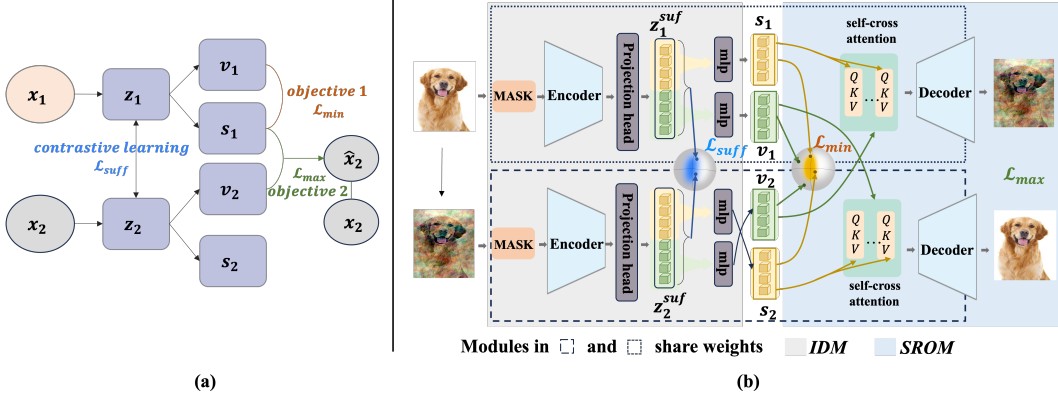

Figure 1: Our method's pipeline, where $x_1$ and $x_2$ denote the input and its augmented counterpart. $z$ represents the sufficient representation, while $s$ and $v$ correspond to the semantic representation and variation. (a) An illustration of our two objectives. First, we employ $\mathcal{L}_{suff}$ to capture sufficient semantics. Subsequently, $\mathcal{L}_{min}$ and $\mathcal{L}_{max}$ are introduced to learn a minimal sufficient representation. We only describe $\mathcal{L}_{min}$ and $\mathcal{L}_{max}$ for $x_1$ in detail. The same procedure applies to $x_2$. (b) A detailed overview of our network and the applied constraints.

Firstly, the InfoNCE loss is applied to constrain $z^{suff}$, which aims to learn sufficient shared information between $x_1$ and $x_2$.

$$\mathcal{L}_{suf} = -\log \frac{exp(z_1^{suff} \cdot z_2^{suff} / \tau)}{\sum_{i=0}^{K} exp(z_1^{suff} \cdot z_k^{suff-} / \tau)}, \tag{1}$$

where $\tau$ is a temperature hyper-parameter and the K is the number of negative samples. $z_k^{suff-}$ represents a negative representation from the negative sample. We construct positive pairs by Fourier-based augmentations (FA) [50], whose efficiency has been experimentally proved in UDG methods [38, 51]. FA randomly alters the amplitudes of one image (anchor) with the amplitudes from another image (target). For our training, the target image is randomly selected from the dataset.

Then, based on the sufficient semantic representation, we introduce two components.

**IDM.** Following intuitive relation $\mathcal{Z} = \mathcal{S} \oplus \mathcal{V}$, we assume that the representation $z$ can be disentangled into a semantic-relevant representation $s$ and a semantic-irrelevant representation $v$. After a shared encoder extracts image features, the features are processed through a projection head to obtain a high-dimensional representation. Subsequently, two MLP modules are applied to $z$ to separate semantic information from variation.

**SROM.** Corresponding to Proposition 3.6, we introduce a semantics and variation mixed InfoNCE $L_{min}$ loss to minimize $I(s; v)$ and a translated reconstruction loss $L_{max}$ to maximize $I(v; x | S)$.

$$\mathcal{L}_{min} = -\log \frac{\exp(s_1 \cdot s_2)}{\sum\limits_{k=0}^{K} \exp(s_1 \cdot s_k^-) + \sum\limits_{k=0}^{K} \exp(s_1 \cdot v_k^-) + \exp(s_1 \cdot v_1)}, \tag{2}$$

where $v_1$ is considered as a negative representation. $\mathcal{L}_{min}$ is modified from InfoNCE loss, which not only maximizes the $I(s_1; s_2)$ but also minimizes $I(s_1; v_1)$.

**Proposition 4.1.** $\mathcal{L}_{min} \geq I(s_1; v_1) - I(s_1; s_2)$

According to Proposition 4.1, minimizing $\mathcal{L}_{min}$ indicates decreasing $I(s_1; v_1)$ and increasing $I(s_1; s_2)$ synchronously. Minimizing $\mathcal{L}_{min}$ satisfies our target to minimize $I(s; v)$. Meanwhile, same as InfoNCE, $\mathcal{L}_{min}$ also keeps $s_1$ as a sufficient semantic representation by maximizing $I(s_1; s_2)$.

$$\mathcal{L}_{max} = \frac{\|\mathbf{D}(s_1, v_2) - x_2\|_2^2 + \|\mathbf{D}(s_2, v_1) - x_1\|_2^2}{2}. \tag{3}$$

Here, $\mathbf{D}$ is the decoder to reconstruct images. Without loss of generality, we consider the object $\mathbf{D}(s_1, v_2)$ to illustrate how $\mathcal{L}_{max}$ maximize $I(v; x | S)$ in the following.

**Proposition 4.2.** $I(v_2, s_1; x_2) - I(x_1; x_2) \leq I(v_2; x_2 | S)$

Proposition 4.2 indicates that $I(v_2, s_1; x_2)$ is related to the lower-bound of $I(v_2; x_2 | S)$. From $\|\mathbf{D}(s_1, v_2) - x_2\|_2^2$, $\mathcal{L}_{max}$ aims to maximize the mutual information $I(v_2, s_1; x_2)$. Thus, $\mathcal{L}_{max}$ can help increase the mutual information $I(v_2; x_2 | S)$ by raising its lower bound

As discussed above, $\mathcal{L}_{min}$ and $\mathcal{L}_{max}$ meet our learning targets described in Section 3.2. Therefore, the final loss is

$$\mathcal{L} = \mathcal{L}_{suff} + \mathcal{L}_{min} + \mathcal{L}_{max}. \tag{4}$$

### 4.2 Theoretical Analysis of Generalization Upper Bound

As discussed in previous works [1, 4] on the generalization error bound based on the multi-distribution learning theory [3], the upper bound of the target risk can be summarized into three parts [4]: source empirical risk, distribution discrepancy between source and target distributions, and confidence bound. Briefly, it can be presented as $R_{P_{(X_{test}, Y_{test})}}(h) \leq R_{P_{(X_{sup}, Y_{sup})}}(h) + \Delta \approx e_{sup} + \Delta$, where $e_{sup}$ is the downstream error, $\Delta$ is the summation of domain discrepancy and confidence bound in downstream [4]. However, we can not access the state of $\Delta$ in the SSL stage. Thus, we focus on the error in the source domain to minimize the generalization bound in the downstream. In the following, we demonstrate the relation between the minimal sufficient semantic representation and the downstream error.

**Theorem 4.3.** *For representation $z_1^{suf}$ and $z_1^{min}$, their Bayes error rates are $e^{min}$ and $e^{suf}$ respectively. $e^{min}$ has the minimum upper bound compared with all minimal sufficient semantic representations. Specifically, given the downstream task $T$, we have*

$$e^{suf} \leq 1 - \exp[-(H(T) + I(z_1^{suf}; x_1 | S)]$$
$$e^{min} \leq 1 - \exp[-(H(T) + I(z_1^{min}; x_1 | S)]$$

Bayes error rate [12] is the minimum achievable error for any representation-learned classifier. Following the previous work [46, 44], we consider it as the downstream error. Theorem 4.3 indicates that downstream error of sufficient semantic representation $z_1^{suf}$ is upper bounded by $I(z_1^{suf}; x_1 | S)$. According to Definition 3.4, the Bayes error rate of $z^{min}$ has the minimum upper bound. Thus, $z^{min}$ provides the lowest upper bound for the risk $R_{P_{(X_{test}, Y_{test})}}(h)$, which helps to improve the generalization ability of the unsupervised models.

Table 1: Performances of UDG and SSL models on PACS dataset. All models are trained on 3 selected domains and tested on the left domains, which process is repeated for all domains. "avg." represents macro-accuracy. We report the accuracy for every domain and the average accuracy for all domains. **Best** and second best are highlighted.

| Methods | Label Fraction: 1% | | | | | Label Fraction: 5% | | | | |
|---|---|---|---|---|---|---|---|---|---|---|
| | Target domain | | | | | Target domain | | | | |
| | *photo* | *art* | *cartoon* | *sketch* | *avg.* | *photo* | *art* | *cartoon* | *sketch* | *avg.* |
| ERM | 10.90 | 11.21 | 14.33 | 18.83 | 13.82 | 14.15 | 18.67 | 13.37 | 18.34 | 16.13 |
| MoCo V2 | 22.97 | 15.58 | 23.65 | 25.27 | 21.87 | 37.39 | 25.57 | 28.11 | 31.16 | 30.56 |
| SimCLR V2 | 30.94 | 17.43 | 30.16 | 25.20 | 25.93 | 54.67 | 35.92 | 35.31 | 36.84 | 40.68 |
| BYOL | 11.20 | 14.53 | 16.21 | 10.01 | 12.99 | 26.55 | 17.79 | 21.87 | 19.65 | 21.46 |
| AdCo | 26.13 | 17.11 | 22.96 | 23.37 | 22.39 | 37.65 | 28.21 | 28.52 | 30.35 | 31.18 |
| MAE | 30.72 | 23.54 | 20.78 | 24.52 | 24.89 | 32.69 | 24.61 | 27.35 | 30.44 | 28.77 |
| DARLING | 27.78 | 19.82 | 27.51 | 29.54 | 26.16 | 44.61 | 39.25 | 36.41 | 36.53 | 39.20 |
| DiMAE | 48.86 | 31.73 | 25.83 | 32.50 | 34.73 | 50.00 | 41.25 | 34.40 | 38.00 | 40.91 |
| BrAD | **61.81** | 33.57 | 43.47 | 36.37 | 43.80 | 65.22 | 41.35 | 50.88 | 50.68 | 52.03 |
| CycleMAE | 52.63 | 36.25 | 35.53 | 34.85 | 39.82 | 63.24 | 39.96 | 42.15 | 36.35 | 45.43 |
| BSS/SimCLR | 43.31 | 38.96 | **48.61** | **48.76** | 44.91 | 58.16 | 46.37 | **55.69** | **65.63** | 56.40 |
| **MS-UDG** | 42.74 | **47.13** | 47.84 | 43.33 | **45.26** | **74.03** | **61.78** | 54.72 | 61.53 | **63.02** |
| Methods | Label Fraction: 10% | | | | | Label Fraction: 100% | | | | |
| | Target Domain | | | | | Target Domain | | | | |
| | *photo* | *art* | *cartoon* | *sketch* | *avg.* | *photo* | *art* | *cartoon* | *sketch* | *avg.* |
| ERM | 16.27 | 16.62 | 18.40 | 12.01 | 15.82 | 43.29 | 24.27 | 32.62 | 20.84 | 30.26 |
| MoCo V2 | 44.19 | 25.85 | 35.53 | 24.97 | 32.64 | 59.86 | 28.58 | 48.89 | 34.79 | 43.03 |
| SimCLR V2 | 54.65 | 37.65 | 46.00 | 28.25 | 41.64 | 67.45 | 43.6 | 54.48 | 34.73 | 50.06 |
| BYOL | 27.01 | 25.94 | 20.98 | 19.69 | 23.40 | 41.42 | 23.73 | 30.02 | 18.78 | 28.49 |
| AdCo | 46.51 | 30.31 | 31.45 | 22.96 | 32.81 | 58.59 | 29.81 | 50.19 | 30.45 | 42.26 |
| MAE | 35.89 | 25.59 | 33.28 | 32.39 | 31.79 | 36.84 | 25.24 | 32.25 | 34.45 | 32.20 |
| DARLING | 53.37 | 39.91 | 46.41 | 30.17 | 42.46 | 68.86 | 41.53 | 56.89 | 37.51 | 51.20 |
| DiMAE | **77.87** | 59.77 | 57.72 | 39.25 | 58.65 | 78.99 | 63.23 | 59.44 | 55.89 | 64.39 |
| BrAD | 72.17 | 44.20 | 50.01 | 55.66 | 55.51 | - | - | - | - | - |
| CycleMAE | 85.94 | **67.93** | 59.34 | 38.25 | 62.87 | **90.72** | **75.43** | 69.33 | 50.24 | 71.41 |
| BSS/SimCLR | 63.29 | 51.37 | 59.43 | 66.09 | 60.04 | 79.50 | 62.73 | 65.67 | **73.02** | 70.23 |
| **MS-UDG** | 74.10 | 61.90 | **63.73** | **73.66** | **68.35** | 84.80 | 63.74 | **71.73** | 71.27 | **72.89** |

# 5 Experiments

## 5.1 Experimental Setup

**Datasets.** Following previous UDG work [56], PACS [23] and DomainNet [35] are evaluated for benchmarking UDG methods. Meanwhile, we also evaluate our methods on OfficeHome, Office31, and VLCS whose results and details can be found in the Appendix.

DomainNet has 345 categories and 6 different domains (Clipart, Infograph, Quickdraw, Painting, Real, and Sketch). PACS has 9,991 images with 7 classes from 4 domains: Art, Cartoons, Photos, and Sketches. For DomainNet, the six domains are split into two sets: 1. Clipart, Infograph, and Quickdraw; 2. Painting, Real, and Sketch as previous work did. 20 classes are selected as both unlabeled and labeled data. We use one set for seen domains and the other one for unseen domains. We report the micro-accuracy and macro-accuracy on six domains. For PACS, three domains are selected for training, and the remaining domain is used for evaluation.

**Experimental Protocol.** We adopt the all-correlated setting, as proposed in DARLING [56] and DisMAE [54]. The overall process is divided into three main steps. First, a pre-trained model is obtained using UDG methods on the unlabeled data from the source domain. Second, this pre-trained model is fine-tuned on varying proportions of labeled data from the source domain, with either the classifier or the entire backbone network being adjusted. Finally, the model is tested on the unseen domain. Following the benchmark [56] and general setting of SSL (i.e., linear probing and fine-tuning), we adopt linear probing for 1% and 5% label fractions and fine-tuning for 10% and 100% label fractions in downstream tasks.

To ensure a fair comparison with previous SSL methods, including MoCo V2 [7], SimCLR V2 [5], BYOL [6], AdCo [19], and MAE [17], as well as UDG methods such as DARLING [56], DiMAE [52], BrAD [16], BSS [39], and CycleMAE [51], the SSL models are initialized using ImageNet pre-

Table 2: Performances of UDG and SSL models on the DomainNet subset, which contains six domains. The models are trained on source domains and evaluated on target domains. "Overall" and "avg." refer to micro-accuracy and macro-accuracy. **Best** and second best are highlighted.

| Label Fraction: 1% | | | | | | | |
|---|---|---|---|---|---|---|---|
| Source | *painting∪real∪sketch* | | | *clipart∪infograph∪quichdraw* | | | |
| Target | *clipart∪infograph∪quichdraw* | | | *painting∪real∪sketch* | | *overall* | *avg.* |
| BYOL | 6.21 | 3.48 | 4.27 | 5.00 | 8.47 | 4.42 | 5.61 | 5.31 |
| MoCo V2 | 18.85 | 10.57 | 6.32 | 11.38 | 14.97 | 15.28 | 12.12 | 12.9 |
| AdCo | 16.16 | 12.26 | 5.65 | 11.13 | 16.53 | 17.19 | 12.47 | 13.15 |
| SimCLR V2 | 23.51 | 15.42 | 5.29 | 20.25 | 17.84 | 18.85 | 15.46 | 16.86 |
| MAE | 22.38 | 12.62 | 10.50 | 17.86 | 24.57 | 19.33 | 17.57 | 17.88 |
| DARLING | 18.53 | 10.62 | 12.65 | 14.45 | 21.68 | 21.30 | 16.56 | 16.54 |
| DiMAE | 26.52 | 15.47 | 15.47 | 20.18 | 30.77 | 20.03 | 21.85 | 21.41 |
| BrAD | 47.26 | 16.89 | 23.74 | 20.03 | 25.08 | 31.67 | 25.85 | 27.45 |
| CycleMAE | 37.54 | 18.01 | 17.13 | 22.85 | 30.38 | 22.31 | 24.08 | 24.71 |
| BSS/SimCLR | **61.94** | 19.58 | **26.98** | 27.40 | 31.55 | 41.49 | 32.27 | 34.82 |
| **MS-UDG** | 55.88 | **20.03** | 18.65 | **32.15** | **39.81** | **42.89** | **32.45** | **34.92** |
| Label Fraction: 5% | | | | | | | |
| Source | *painting∪real∪sketch* | | | *clipart∪infograph∪quichdraw* | | | |
| Target | *clipart∪infograph∪quichdraw* | | | *painting∪real∪sketch* | | *overall* | *avg.* |
| BYOL | 9.60 | 5.09 | 6.02 | 9.78 | 10.73 | 3.97 | 7.83 | 7.53 |
| MoCo V2 | 28.13 | 13.79 | 9.67 | 20.8 | 24.91 | 21.44 | 18.99 | 19.79 |
| AdCo | 30.77 | 18.65 | 7.75 | 19.97 | 24.31 | 24.19 | 19.42 | 20.94 |
| SimCLR V2 | 34.03 | 17.17 | 10.88 | 21.35 | 24.34 | 27.46 | 20.89 | 22.54 |
| MAE | 32.6 | 15.28 | 13.43 | 24.55 | 30.43 | 26.07 | 22.88 | 23.73 |
| DARLING | 39.32 | 19.09 | 10.50 | 21.09 | 30.51 | 28.49 | 23.31 | 24.83 |
| DiMAE | 42.31 | 18.87 | 15.00 | 27.02 | 39.92 | 26.50 | 27.85 | 28.27 |
| BrAD | 37.54 | 18.01 | 17.13 | 22.85 | 30.38 | 22.31 | 24.08 | 24.71 |
| CycleMAE | 55.14 | 20.87 | 19.62 | 27.64 | 40.24 | 28.71 | 30.80 | 32.04 |
| BSS/SimCLR | **71.21** | 20.93 | **32.42** | 36.68 | 41.49 | **52.75** | 39.73 | 42.58 |
| **MS-UDG** | 71.02 | **28.21** | 28.51 | **40.07** | **47.05** | 49.51 | **41.33** | **44.06** |
| Label Fraction: 10% | | | | | | | |
| Source | *painting∪real∪sketch* | | | *clipart∪infograph∪quichdraw* | | | |
| Target | *clipart∪infograph∪quichdraw* | | | *painting∪real∪sketch* | | *overall* | *avg.* |
| BYOL | 14.55 | 8.71 | 5.92 | 9.50 | 10.38 | 4.45 | 8.69 | 8.92 |
| MoCo V2 | 32.46 | 18.54 | 8.05 | 25.35 | 29.91 | 23.71 | 21.87 | 23.00 |
| AdCo | 32.25 | 17.96 | 11.56 | 23.35 | 29.98 | 27.57 | 22.79 | 23.78 |
| SimCLR V2 | 37.11 | 19.87 | 12.33 | 24.01 | 30.17 | 31.58 | 24.28 | 25.84 |
| MAE | 51.86 | 24.81 | 23.94 | 41.24 | 54.68 | 39.41 | 38.85 | 39.32 |
| DARLING | 35.15 | 20.88 | 15.69 | 25.90 | 33.29 | 30.77 | 26.09 | 26.95 |
| DiMAE | 70.78 | 38.06 | 27.39 | 50.73 | 64.89 | 55.41 | 49.49 | 51.21 |
| BrAD | 68.27 | 26.60 | 34.03 | 31.08 | 38.48 | 48.17 | 38.74 | 41.10 |
| CycleMAE | 74.87 | **38.42** | 28.32 | 52.81 | **67.13** | 56.37 | 50.78 | 52.98 |
| BSS/SimCLR | 71.95 | 21.27 | 33.47 | 39.49 | 44.67 | 55.42 | 41.57 | 44.38 |
| **MS-UDG** | **79.70** | 30.01 | **40.11** | **53.73** | 63.77 | **65.82** | **53.37** | **55.52** |

trained model for the DomainNet and PACS datasets. Additionally, SSL training without ImageNet pre-trained model is conducted for comparison with DisMAE [54] on DomainNet. Further details of this experiment can be found in the Appendix.

**Implementation Details.** Following previous disentanglement-based UDG methods [51, 52], we adopt ViT-S/16 [9] as the backbone. The learning rate for pre-training is set to $1 \times 10^{-4}$. The weight decay is set to 0.05, and the batch size is 32. For $\tau$ in $L_{suff}$, we set the temperature $\tau = 0.07$ as previous work [18]. During fine-tuning, all methods are trained for 50 epochs, and the best evaluation model is selected to test on unseen domain data, following the exact training schedule outlined in [54]. The test domains also remain unseen during the model selection. Further details can be found in the Appendix.

## 5.2 Experimental Results

We present the experimental results in Tab. 1 (PACS) and Tab. 2 (DomainNet). Compared to contrastive-based and generative-based SSL methods MoCo V2, SimCLR V2, BYOL, AdCo, and MAE, MS-UDG outperforms in most cases. Among contrastive-based methods, SimCLR v2, which

uses multi-view augmented images as self-supervised signals, performs best. Compared with other SSL methods, MAE demonstrates better performance on the DomainNet dataset, second only to SimCLR V2 on the PACS dataset. From the results, these SSL methods perform poorly on downstream tasks with unseen domain data, as they ignore domain covariates.

Compared to the state-of-the-art (SOTA) SSL method SimCLR V2, MS-UDG improves the performance by **+19.33%**, **+22.34%**,**+26.71%**, and **+22.83%** for label fractions 1%, 5%, 10% and 100%, respectively, on the PACS dataset. On the DomainNet dataset, MS-UDG improves the SOTA SSL method MAE by **+17.04%**, **+18.45%**, and **+14.52%** in overall accuracy for label fractions 1%, 5%, and 10%.

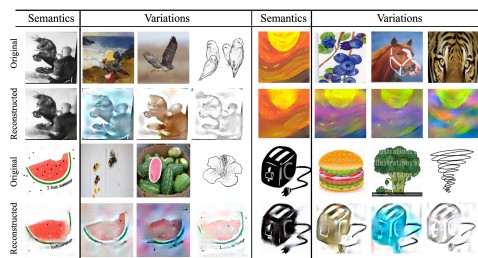

From our results, MS-UDG outperforms most contrastive-based and disentanglement-based UDG methods across two datasets. Specifically, in linear evaluation with 1% and 5% label fractions, MS-UDG outperforms the SOTA method, BSS, by **+0.35%** and **+6.62%**, respectively, on the PACS dataset. On the DomainNet dataset, MS-UDG improves by **+0.18%** and **+1.6%** in linear evaluation compared to BSS. In full fine-tuning with 10% and 100% label fractions, MS-UDG surpasses the SOTA method CycleMAE by **+5.48%** and **+1.48%**, respectively, on the PACS dataset. On the DomainNet dataset, MS-UDG shows an improvement of **+2.59%** overall accuracy for the 10% label fraction.

Figure 2: The reconstruction results produced by MS-UDG. Rows 1 and 3 display input images that preserve either semantic content or stylistic variations, derived from four distinct domains within the DomainNet dataset. In contrast, Rows 2 and 4 present reconstructed images using alternative variation representations within the feature space.

### 5.3 Ablation Study

**Effectiveness of Each Component of MS-UDG.** Since MS-UDG is based on two optimization objectives, we investigate the impact of each target. Our baseline is the native InfoNCE loss $\mathcal{L}_{suff}$. Next, we introduce the IDM and SROM modules and evaluate $\mathcal{L}_{min}$ and $\mathcal{L}_{max}$, separately. Finally, we assess the performance of the entire method, incorporating $\mathcal{L}_{suff}$, $\mathcal{L}_{min}$, and $\mathcal{L}_{max}$.

We evaluate the effectiveness on the PACS dataset. From Tab. 3, both applying $\mathcal{L}_{min}$ and $\mathcal{L}_{max}$ improve the baseline, while the performance can be further improved by combining $\mathcal{L}_{min}$ and $\mathcal{L}_{max}$ by **+2.81%**, **6.85%**, and **+3.54%** under 1%, 5%, and 10% label fractions. Combining the two losses further improves the results, aligning with our theoretical analysis. Minimizing $I(s;v)$ and maximizing $I(v;x|S)$ reduce the semantic-irrelevant information $I(s;x|S)$.

Table 3: Effectiveness of each component of MS-UDG on PACS dataset. All models are trained on 3 selected domains and tested on the remaining domains, which process is repeated for all domains. **Best** is highlighted.

| Methods | Label Fraction | | |
|---|---|---|---|
| | 1% | 5% | 10% |
| Baseline ($\mathcal{L}_{suff}$) | 42.45 | 56.17 | 64.81 |
| +$\mathcal{L}_{min}$ | 42.85 | 57.91 | 65.32 |
| +$\mathcal{L}_{max}$ | 44.77 | 57.72 | 67.40 |
| +$\mathcal{L}_{min}$+$\mathcal{L}_{max}$ | **45.26** | **63.02** | **68.35** |

**Visualization of Reconstruction.** Fig. 2 illustrates reconstructed images with consistent semantics and varying styles, showcasing MS-UDG's strong capability to separate features.

## 6 Conclusion and Limitation

In conclusion, we tackle the challenge of semantic disentanglement in UDG without domain labels by formulating it as a constrained optimization problem. Our framework, MS-UDG, integrates contrastive learning with novel modules to effectively separate semantics and variation, achieving state-of-the-art performance. However, MS-UDG is primarily tailored for scenarios with pronounced domain discrepancies. This assumption may limit its effectiveness in settings with subtle covariate shifts, where improvements in disentanglement are less evident.

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

# A Explanation of Concepts

## A.1 Minimal Sufficient Representation

**Minimal Sufficient Statistic:** The minimal sufficient statistic (MSS) proposes a concept of effective statistic [8]. Given two sets of data $X, Y$, $(X, Y) \sim P(X, Y)$, $F \triangleq f_\theta(X)$ is a statistic of $X$, if the Markov chain $X \to F \to Y$ is satisfied, we can define the sufficient statistic, which captures the all information between $X$ and $Y$.

**Definition A.1** (Sufficient Statistic [8]). $F$ is a sufficient statistic for $Y$ if and only if $I(F, Y) = I(X; Y)$

However, the definition of sufficient statistic can not effectively extract the essential information [40]. For instance, directly 'copy' can satisfy the above definition. Thus, the minimal sufficient statistic is introduced.

**Definition A.2** (Minimal Sufficient Statistic). A sufficient statistic $F$ is minimal sufficient if and only if, any sufficient statistic $S$, $\exists g$ as a function, $F = g(S)$

The MSS are sufficient statistics containing the least information about $X$ [22]. Based on the property of MSS, many methods are proposed to obtain the effective representations, such as Information Bottleneck [43]

**Minimal Sufficient Representation:** Motivated from the above discussion, the minimal sufficient representation is proposed for self-supervised learning [44, 46]. Definition 3.1 introduce the definition of minimal sufficient representation, which is similar to the definition of minimal sufficient statistic. However, the minimal sufficient representation can still capture redundant information, especially when large distribution discrepancies exist in the dataset. For instance, the minimal sufficient representations from sketch images in PACS dataset may still capture the information of sketch domain. Therefore, we propose the definition of minimal sufficient semantic representation to handle this issue.

## A.2 OOD Generalization Bound

From the probably approximately correct (PAC) learning theory on multi-distribution [3], the out-of-ditribution generalization bound has been proposed [1, 4]

**Theorem A.3.** *Consider the risk on source domain $D_i$ is $R_{P_{(X_{D_i}, Y_{D_i})}}(h)$, then, we have*

$$R_{P_{(X_{test}, Y_{test})}}(h) \leq \sum_i^{N_d} R_{P_{(X_{D_i}, Y_{D_i})}}(h) + div(\hat{D}_s, D_t) + \lambda$$

*Where $div(D_s, D_t)$ is the divergence between the source distributions $\hat{D}_s$ and the unseen target distribution $D_t$, $\lambda$ is the confidence bound, and $N_d$ is the number of source domains.*

For $div(\hat{D}_s, D_t)$, Albuquerque *et al.*[1] define it as the distance from unseen distribution $D_t$ to the simplex supported by the source distributions $\hat{D}_s$, Cha *et al.* [4] simply define it as $\frac{1}{2N_d} \sum_i^{N_d} 2 \sup_A |P_{D_i}(A) - P_{D_t}(A)|$. For $\lambda$, Albuquerque *et al.*[1] consider it as the shift of the labeling function on the unseen distribution, and Cha *et al.* [4] relate it to the VC dimension of the model. In our paper, as we can not control these two items, we simply use $\Delta$ to present the summation of them. In addition, we intuitively consider the source risk as $R_{P_{(X_{sup}, Y_{sup})}}(h) = \sum_i^{N_d} R_{P_{(X_{D_i}, Y_{D_i})}}(h)$. Therefore, we can utilize Theorem 4.3 to approximate the upper bound of the target risk $R_{P_{(X_{test}, Y_{test})}}(h)$

# B Theoretical Proof

***Proof of Proposition 3.2.*** $I(x_i; T; \hat{z}_1^{min}) = I(\hat{z}_1^{min}; T) - I(\hat{z}_1^{min}; T | x_i)$. With lemma 1 in [46], $I(\hat{z}_1, T | x_1)$ and $I(\hat{z}_1, T | x_2)$ should be zero. By the definition of interaction information, we have

$$I(x_i; \hat{z}_1^{min}) = I(x_i; T; \hat{z}_1^{min}) + I(x_i; \hat{z}_1^{min} | T)$$
$$= I(\hat{z}_1^{min}; T) + I(x_i; \hat{z}_1^{min} | T)) \geq I(\hat{z}_1^{min}; T)$$

$\square$

**Proof of Proposition 3.5.** Considering the Markov chain $x_1 \rightarrow z_1$, $\forall f$, $f$ is arbitrary conditional factor, we have $I(z_1; f|x_1) = 0$. This also the assumption in [46], so we have

$$I(z_1; x_1, x_2|S) = I(z_1; x_1|S) + I(z_1; x_2|S, x_1)$$
$$= I(z_1; x_1|S)$$

$\square$

**Proof of Proposition 3.6.** Following the assumption in Proof of Proposition 3.5, we have $I(s; v|x) = 0$. Then

$$I(z; x|S) = I(s, v; x|S)$$
$$= I(s; x|S) + I(v; x|S, s)$$
$$= I(s; x|S) + I(v; x|S) - I(v; x; s|S)$$
$$= I(s; x|S) + I(v; x|S) - I(s; v|S) + I(s; v|S, x)$$
$$= I(s; x|S) + I(v; x|S) - I(s; v|S)$$

Since $I(s; v) = I(s; v|S)$, we have

$$I(s; x|S) = I(z; x|S) + I(s; v) - I(v; x|S)$$

$\square$

**Proof of Proposition 4.1.** According to [32], the log-bilinear model can be represented as a density ratio:

$$f(a, b) = \exp(a, b/\tau) \propto \frac{p(b|a)}{p(b)}$$

$a, b$ are randomly given variables. In addition, The mutual information can be expressed as

$$I(a; b) = \mathbb{E}_X \log(\frac{p(b|a)}{p(b)})$$

$X$ is the given dataset. We set $Z_{neg}$ are representations from other images instead of $x_1$, which means $s_k^-, v_k^- \in Z_{neg}$, $|Z_{neg}| = 2K$ Then, we have

$$\mathcal{L}_{min} = -\mathbb{E}_{X_{ssl}} \log[\frac{\frac{p(s_2|s_1)}{p(s_2)}}{\frac{p(s_2|s_1)}{p(s_2)} + \sum_{z^- \in Z_{neg}} \frac{p(z^-|s_1)}{p(z^-)} + \frac{p(v_1|s_1)}{p(v_1)}}]$$

$$\approx \mathbb{E}_{X_{ssl}} \log[1 + 2\frac{p(s_2)}{p(s_2|s_1)}K + \frac{p(s_2)}{p(s_2|s_1)}\frac{p(v_1|s_1)}{p(v_1)}] \quad \text{according to [32]}$$

$$\geq \mathbb{E}_{X_{ssl}} \log[\frac{p(s_2)}{p(s_2|s_1)}\frac{p(v_1|s_1)}{p(v_1)}]$$

$$= I(v_1; s_1) - I(s_1; s_2)$$

$\square$

**Proof of Proposition 4.2.** Since $0 = I(v_2; S)$, we have $I(v_2; x_2|s_1) \leq I(v_2; x_2) = I(v_2; x_2|S)$. Thus

$$I(v_2, s_1; x_2) = I(s_1; x_2) + I(v_2; x_2|s_1)$$
$$\leq I(s_1; x_2) + I(v_2; x_2|S)$$

From our assumption, $s_1$ may not contain all the information in the shared information between $x_1$ and $x_2$, thus $I(s_1; x_2|S) \leq I(x_1; x_2|S)$. Then, with Definition 3.3, we have

$$I(s_1; x_2) = I(s_1; x_2; S) + I(s_1; x_2|S))$$
$$\leq I(x_1; x_2; S) + I(x_1; x_2|S)$$
$$= I(x_1; x_2)$$

Then,

$$I(v_2, s_1; x_2) - I(x_1; x_2) \leq I(v_2, s_1; x_2) - I(s_1; x_2) \leq I(v_2; x_2 | S)$$

$\square$

**Proof of theorem 4.3.** According to [10], the equality between $e^{suf}$ and $H(T | z_1^{suf})$ is

$$-ln(1 - e^{suf}) \leq H(T | z_1^{suf})$$

Generally, we consider that $X_{ssl}$ is large enough to cover the semantic information in $X_{sup}$, thus we have $I(z_1^{suf}; S) = I(z_1^{suf}; T)$. Then,

$$H(T | z_1^{suf}) = H(T) - H(S) + H(S | z_1^{suf})$$

So, we should consider the $H(S | z_1^{suf})$ for the upper bound of $e^{suf}$

$$
\begin{aligned}
H(S | z_1^{suf}) &= H(S | x_1, z_1^{suf}) + I(x_1; S | z_1^{suf}) \\
&= H(S | x_1) + I(x_1; S | z_1^{suf}) \\
&= H(S | x_1) + I(x_1; z_1^{suf} | S) - I(x_1; z_1^{suf}) + I(x_1; S) \\
&\leq H(S | x_1) + I(x_1; z_1^{suf} | S) + H(x_1) - H(x_1 | S) \\
&= H(S) + I(x_1; z_1^{suf} | S)
\end{aligned}
$$

Thus, $H(T | z_1^{suf}) \leq H(T) + I(x_1; z_1^{suf} | S)$. Then we have

$$e^{suf} \leq 1 - exp[-(H(T) + I(z_1^{suf}; x_1 | S))]$$

Since $I(z_1^{min}; x_1 | S) \leq I(z_1^{suf}; x_1 | S)$, the upper bound of $e^{min}$ is minimum in the sufficient representations. $\square$

## C   Implementation Details

For pre-training, we use ViT-S/16 as the backbone and the same for fine-tuning. The pre-training follows a cosine decay schedule. All experiments were conducted on a single Nvidia A100 80GB GPU, an 8-core CPU, and 250GB of memory. The details of experimental settings can be seen in Tab. 4. For all models, the input size is $224 \times 224$. When performing linear probing, we adopt the same normalization scheme as described in [17] by incorporating a batch normalization layer. Experiments are conducted three times with different random seeds, and the average accuracy from these runs is reported.

Table 4: Experimental Settings. Lp represents linear probing, and ft represents fine-tuning.

| label fraction | strategy | backbone | epoch | batch_size | weight_decay | learning_rate | optimizer | betas |
|---|---|---|---|---|---|---|---|---|
| Pretraining (DomainNet and PACS) | | | | | | | | |
| - | - | ViT-S/16 | 80 | 192 | 0.05 | 1e-4 | AdamW | (0.9,0.95) |
| Finetuning (DomainNet) | | | | | | | | |
| 1% | lp | ViT-S/16 | 50 | 32 | 0.05 | 5e-4 | AdamW | (0.9,0.95) |
| 5% | lp | ViT-S/16 | 50 | 128 | 0.05 | 5e-4 | AdamW | (0.9,0.95) |
| 10% | ft | ViT-S/16 | 50 | 128 | 0.05 | 5e-4 | AdamW | (0.9,0.95) |
| 100% | ft | ViT-S/16 | 50 | 128 | 0.05 | 5e-4 | AdamW | (0.9,0.95) |
| Finetuning (PACS) | | | | | | | | |
| 1% | lp | ViT-S/16 | 50 | 16 | 0.05 | 5e-4 | AdamW | (0.9,0.95) |
| 5% | lp | ViT-S/16 | 50 | 64 | 0.05 | 5e-4 | AdamW | (0.9,0.95) |
| 10% | ft | ViT-S/16 | 50 | 64 | 0.05 | 5e-5 | AdamW | (0.9,0.95) |
| 100% | ft | ViT-S/16 | 50 | 64 | 0.05 | 5e-5 | AdamW | (0.9,0.95) |

## D   Effects of Fourier-based Augmentation

By comparing Tab. 3 with Tab. 1, we observe that the baseline, *i.e.*, using only $\mathcal{L}_{suff}$, outperforms certain UDG methods. Prior work [56] has shown that Fourier-based Augmentation (FA) enhances

UDG performance. Here, we further investigate its impact on our method. As presented in Tab. 5, $\mathcal{L}_{suff}$ without FA corresponds to the native InfoNCE loss, aligning with MoCo V2 [7], and achieves comparable average performance. However, incorporating FA significantly boosts the baseline by +21.22%. Our findings are consistent with BSS [39], where FA improves SimCLR V2's baseline performance by +16.02%, further underscoring the effectiveness of FA in self-supervised learning.

Table 5: Results of detailed ablation study on PACS in label fraction 1%. "w/o FA" and "w/ FA" denote training without and with Fourier-based Augmentation, respectively. **Best** is highlighted.

| Methods | photo | art | cartoon | sketch | avg. |
|---|---|---|---|---|---|
| MoCo V2 | 22.97 | 15.58 | 23.65 | 25.27 | 21.87 |
| Baseline ($\mathcal{L}_{suff}$ w/o FA) | 25.02 | 23.75 | 16.85 | 19.31 | 21.23 |
| Baseline ($\mathcal{L}_{suff}$ w/ FA) | 34.89 | 45.9 | 47.68 | 41.33 | 42.45 |
| $+\mathcal{L}_{min}$ | 35.55 | 44.93 | 47.84 | 43.08 | 42.85 |
| $+\mathcal{L}_{max}$ | 39.82 | 46.28 | 47.04 | 45.94 | 44.77 |
| $+\mathcal{L}_{min}+\mathcal{L}_{max}$ | **42.74** | **47.13** | **47.84** | **43.33** | **45.26** |

# E    Learning without ImageNet Pre-trained Model

In this section, we discuss the performance of UDG methods learning without initializing with a pre-trained model on ImageNet. To compare with previous work [54] fairly, we use ViT-B/16 as our backbone, and the depth of the decoder layer is 1, following the setting in the work [54]. As shown in Tab. 6, MS-UDG outperforms SOTA methods in most settings.

# F    Effects of Decoder Depth

As previous studies explored the impact of decoder depth (the number of multi-head attention blocks, as in MAE), we present the results for different decoder depths on DomainNet in Tab. 7. In line with prior work [49, 51], our findings confirm that a lightweight decoder achieves superior performance.

# G    Visualization of Extracted Features on Multi-domain Data

We present t-SNE visualization of six domain features from MS-UDG and MAE on the DomainNet datasets. From Fig. 3, MS-UDG separates variations well.

Table 6: Evaluation of UDG and SSL models which train without the initialization from the ImageNet pre-trained model. Most results are drawn from the work [21]. We keep the same setting with DisMAE: pre-training 500 epochs and fine-tuning 50 epochs. **Best** is highlighted.

| | Label Fraction 1% | | | | | Label Fraction 5% | | | | |
|---|---|---|---|---|---|---|---|---|---|---|
| | Target domain paint∪real∪sketch | | | overall | avg. | Target domain paint∪real∪sketch | | | overall | avg. |
| ERM | 7.98 | 9.94 | 5.38 | 8.46 | 7.77 | 6.48 | 8.64 | 9.84 | 8.29 | 8.32 |
| MAE | 8.80 | 10.30 | 12.62 | 10.38 | 10.57 | 12.13 | 17.63 | 15.02 | 15.60 | 14.93 |
| DARLING | 8.59 | 9.01 | 11.10 | 9.32 | 9.57 | 9.31 | 12.00 | 13.72 | 11.61 | 11.68 |
| CycleMAE | 9.21 | 9.49 | 6.62 | 8.82 | 8.44 | 11.44 | 14.21 | 10.01 | 12.58 | 11.88 |
| DisMAE | 11.04 | 11.64 | 12.46 | 11.65 | 11.71 | 13.69 | 17.74 | 16.92 | 16.47 | 16.12 |
| MS-UDG | **12.38** | **12.41** | **12.77** | **12.52** | **12.48** | **16.87** | **21.87** | **17.52** | **19.59** | **18.76** |
| | Label Fraction 10% | | | | | Label Fraction 100% | | | | |
| | Target domain paint∪real∪sketch | | | overall | avg. | Target domain paint∪real∪sketch | | | overall | avg. |
| ERM | 14.14 | 16.76 | 12.63 | 15.19 | 14.51 | 26.25 | 33.29 | 23.25 | 29.28 | 27.59 |
| MAE | 19.34 | 23.07 | 24.18 | 22.29 | 22.20 | 31.72 | 43.72 | 36.74 | 39.02 | 37.40 |
| DARLING | 13.72 | 19.76 | 16.43 | 17.40 | 16.64 | 25.87 | 37.60 | 26.67 | 32.11 | 30.05 |
| CycleMAE | 17.80 | 22.93 | 17.15 | 20.23 | 19.29 | 31.72 | 39.44 | 30.00 | 35.39 | 33.72 |
| DisMAE | **24.49** | 27.06 | **28.24** | 26.61 | **26.60** | 38.89 | 45.95 | 43.72 | 43.58 | 42.85 |
| MS-UDG | 22.27 | **29.31** | 28.02 | **27.09** | 26.53 | **39.82** | **51.60** | **45.08** | **46.99** | **45.50** |

Table 7: The results on the different depths of the decoder layer on DomainNet. The fine-tuning results are conducted on source domain set {clipart, infograph, quickdraw} and target domain set {painting, real, sketch} with a label fraction of 1%. **Best** is highlighted.

| layer depth | overall | avg. |
|:-----------:|:-------:|:----:|
| 1 | **27.63** | **31.02** |
| 2 | 25.83 | 26.63 |
| 4 | 25.33 | 26.55 |
| 8 | 24.05 | 24.94 |

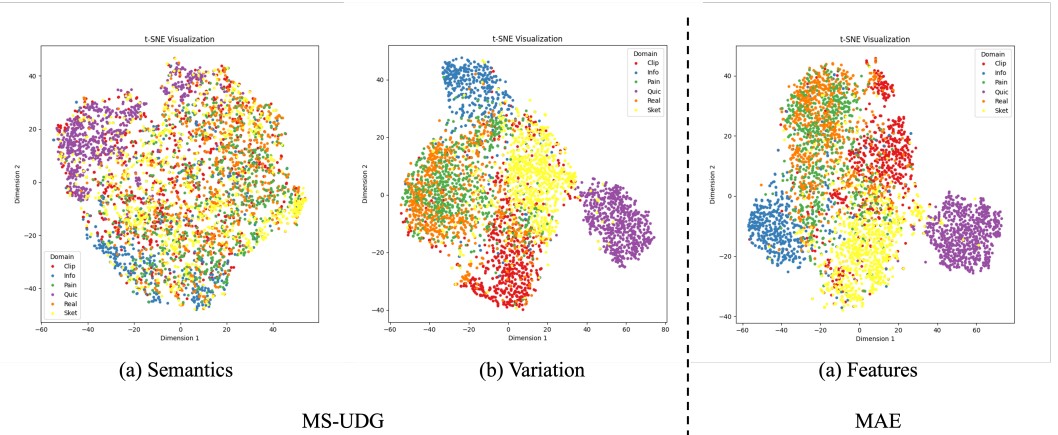

(a) Semantics     (b) Variation     (a) Features

MS-UDG       MAE

Figure 3: We present the t-SNE visualization of six domain features from MS-UDG and MAE on the DomainNet dataset, with semantics and variation features of MS-UDG visualized separately.

## H    Effects of Cross Attention in SROM

In the SROM module, we use the cross attention to fuse $s_1$ and $v_2$ inspired by Stable Diffusion [36], which uses cross attention on prompts and image features. Here, we view the semantic representation $s_1$ as a prompt to fuse semantics and variation. The ablation study of cross-attention is shown in Tab. 8. Using cross attention to fuse $v$ and $s$ further enhances performance.

Table 8: Results with and without cross-attention. "w/o Cross Attn" and "w/ Cross Attn" denote training without and with cross-attention, respectively. **Best** is highlighted.

| | Label Fraction 1% | | | | |
|:---|:---:|:---:|:---:|:---:|:---:|
| Source
Target | painting∪real∪sketch
clipart∪infograph∪quickdraw | | | overall | avg. |
| w/o Cross Attn | **54.82** | 19.47 | 18.94 | 26.36 | **31.08** |
| w/ Cross Attn | 49.44 | **20.90** | **22.71** | **27.63** | 31.02 |
| | Label Fraction 10% | | | | |
| Source
Target | painting∪real∪sketch
clipart∪infograph∪quickdraw | | | overall | avg. |
| w/o Cross Attn | 79.03 | **32.89** | 36.31 | 44.02 | 49.41 |
| w/ Cross Attn | **79.70** | 30.01 | **40.11** | **45.33** | **49.94** |

## I    Results on More Benchmarks

We evaluate several SOTA methods on more benchmarks OfficeHome [45], VLCS [1], and Office31 [37]. As those datasets have limited images, making them less ideal for SSL, we used them as downstream tasks by pre-training on DomainNet and evaluating on other datasets. To preserve

Table 9: Evaluation on OfficeHome. **Best** is highlighted.

| Label Fraction 1% | | | | |
|---|---|---|---|---|
| Pre-training Domain(DomainNet) | Painting∪Real∪Sketch | | | |
| Target Domain(OfficeHome) | Art | Clipart | Product | *avg.* |
| MAE | 13.41 | 24.63 | 23.21 | 20.42 |
| DisMAE | 22.89 | 18.07 | 43.79 | 28.25 |
| BSS | 24.39 | 27.16 | 35.53 | 29.03 |
| MS-UDG | 27.71 | 24.79 | 37.71 | **30.07** |
| Label Fraction 5% | | | | |
| Pre-training Domain(DomainNet) | Painting∪Real∪Sketch | | | |
| Target Domain(OfficeHome) | Art | Clipart | Product | *avg.* |
| MAE | 42.68 | 43.79 | 42.41 | 42.96 |
| DisMAE | 44.64 | 45.82 | 45.2 | 45.22 |
| BSS | 43.9 | 50.53 | 42.69 | 45.71 |
| MS-UDG | 49.4 | 58.16 | 56.78 | **54.78** |

the unseen domain setting, overlapping domains were excluded. Those benchmarks are all adopted leave-one-domain-out validation. The results, compared with SOTA UDG methods DisMAE and BSS, are summarized in Tab. 9, 10, and 11. MS-UDG outperforms in those benchmarks as well.

OfficeHome consists of four domains with 15,500 images: Art, Clipart, Product, and Real-World. The Real-World domain is excluded from the evaluation because of the overlap with the pre-training domain of DomainNet. VLCS consists of images from four distinct domains with around 10,000 images: Caltech101, LabelMe, SUN09, and VOC2007. Office31 consists of images from three distinct domains with 4110 images: Amazon, Webcam, and DSL. More details of datasets can be found in DomainBed [14].

## J   A Comparison of UDG and DG

Given the differences in scope and the available prior knowledge between DG and UDG, as discussed in the following paragraph, making a direct and fair comparison between the two proves challenging. To further explore this, we evaluate the performance of combining UDG with DG, i.e., pre-training with UDG methods followed by supervised learning using DG methods. We conducted some experiments with the UDG+DG approach as shown in Tab. 12. We observed some interesting results. When using 1% labeled data, SWAD negatively impacted the performance of all UDG models. However, with 100% labeled data, linear probing and fine-tuning produced different outcomes. Fine-tuning with SWAD improved the performance of DisMAE and MS-UDG, while slightly degrading the results for BSS and MAE. On the other hand, linear probing with SWAD led to slight performance degradation across all methods, except for MAE. This could be due to the limited number of tuned parameters in linear probing or the small training data in few-shot settings, which might make it difficult for SWAD to find flat minima. This is an intriguing phenomenon and presents an interesting avenue for future research in this area.

In the following, we discuss the detailed differences between UDG and DG. Unlike traditional DG methods, UDG leverages unlabeled data through a self-supervised learning strategy to improve generalization on downstream tasks. In other words, UDG provides an unsupervised pre-trained model (upstream), whereas traditional DG methods train a supervised model (downstream). Below are the differences from the setup and theory perspectives.

Firstly, from a setup perspective, traditional DG methods typically train on category labels to enhance generalization, whereas UDG uses only unlabeled data to train a pre-trained model. Given the differences in scope and the available prior knowledge between DG and UDG, we believe it is challenging to make a direct and fair comparison between the two.

Secondly, from the theory perspective, traditional DG methods always focus on the generalization bound of the target risk since the methods are built on the available category labels. Thus, the original theory of DG is hard to adapt to the UDG area.

Table 10: Evaluation on VLCS. **Best** is highlighted.

| Label Fraction 1% | | | | | |
|---|---|---|---|---|---|
| Pre-training Domain(DomainNet) | Clipart∪Infograph∪Quickdraw | | | | |
| Target Domain(VLCS) | Caltech101 | LabelMe | SUN09 | VOC2007 | avg. |
| MAE | 62.79 | 46.85 | 39.19 | 43.93 | 48.19 |
| DisMAE | 68.17 | 57.24 | 41.66 | 45.74 | **53.20** |
| BSS | 58.20 | 44.31 | 40.25 | 42.04 | 46.20 |
| MS-UDG | 68.17 | 49.11 | 41.93 | 45.04 | 51.06 |
| Label Fraction 5% | | | | | |
| Pre-training Domain(DomainNet) | Clipart∪Infograph∪Quickdraw | | | | |
| Target Domain(VLCS) | Caltech101 | LabelMe | SUN09 | VOC2007 | avg. |
| MAE | 69.84 | 49.01 | 44.44 | 43.93 | 51.81 |
| DisMAE | 69.93 | 54.23 | 43.37 | 43.22 | 52.69 |
| BSS | 68.34 | 47.13 | 40.21 | 41.6 | 49.32 |
| MS-UDG | 79.01 | 52.12 | 54.65 | 49.78 | **58.89** |

Table 11: Evaluation on Office31. **Best** is highlighted.

| Label Fraction 1% | | | | |
|---|---|---|---|---|
| Pre-training Domain(DomainNet) | Clipart∪Infograph∪Quickdraw | | | |
| Target Domain(Office31) | Amazon | DSLR | Webcam | *avg.* |
| MAE | 21.05 | 20.00 | 20.63 | 20.56 |
| DisMAE | 21.00 | 30.77 | 20.63 | 24.13 |
| BSS | 22.81 | 18.46 | 17.46 | 19.58 |
| MS-UDG | 24.56 | 41.54 | 19.05 | **28.38** |
| Label Fraction 5% | | | | |
| Pre-training Domain(DomainNet) | Clipart∪Infograph∪Quickdraw | | | |
| Target Domain(Office31) | Amazon | DSLR | Webcam | *avg.* |
| MAE | 20.8 | 32.31 | 27.78 | 26.96 |
| DisMAE | 45.11 | 33.85 | 33.33 | 37.43 |
| BSS | 44.86 | 35.38 | 34.13 | 38.12 |
| MS-UDG | 58.40 | 52.31 | 50.00 | **53.57** |

# K   Broader Impacts

This research seeks to advance the field of Unsupervised Domain Generalization (UDG) by addressing two core objectives: disentangling semantics and variation without domain labels and category labels. These efforts have the potential to make significant contributions across multiple domains, including healthcare, where the ability to generalize from limited labeled data is of paramount importance. In fields where large-scale unlabeled data is scarce but robust generalization is essential, our approach to disentangling these factors in an unsupervised manner could provide a critical insight, enabling models to perform well in unseen environments without requiring extensive labeled data.

Table 12: Evaluation on UDG+DG(SWAD) methods.

| Label Fraction 1% (Linear probing) | | | | |
|---|---|---|---|---|
| Pre-training Domain(DomainNet) | Painting∪Real∪Sketch | | | |
| Target Domain(OfficeHome) | Art | Clipart | Product | *avg.* |
| MAE | 13.41 | 24.63 | 23.21 | 20.42 |
| MAE+SWAD | 10.84 | 17.23 | 21.14 | 16.40 |
| DisMAE | 22.89 | 18.07 | 43.79 | 28.25 |
| DisMAE+SWAD | 12.05 | 18.28 | 23.71 | 18.01 |
| BSS | 24.39 | 27.16 | 35.53 | 29.03 |
| BSS+SWAD | 13.86 | 23.95 | 27.14 | 21.65 |
| MS-UDG | 27.71 | 24.79 | 37.71 | 30.07 |
| MS-UDG+SWAD | 21.08 | 26.89 | 38.29 | 28.75 |
| Label Fraction 100%(Fine tuning) | | | | |
| Pre-training Domain(DomainNet) | Painting∪Real∪Sketch | | | |
| Target Domain(OfficeHome) | Art | Clipart | Product | *avg.* |
| MAE | 54.00 | 67.53 | 69.31 | 63.61 |
| MAE+SWAD | 48.00 | 67.13 | 71.43 | 62.19 |
| DisMAE | 55.50 | 67.13 | 75.4 | 66.01 |
| DisMAE+SWAD | 59.00 | 74.3 | 76.98 | 70.09 |
| BSS | 51.00 | 63.35 | 65.87 | 60.07 |
| BSS+SWAD | 47.5 | 63.15 | 65.61 | 58.75 |
| MS-UDG | 59.00 | 69.72 | 73.28 | 67.33 |
| MS-UDG+SWAD | 62.5 | 73.51 | 76.46 | 70.82 |
| Label Fraction 100%(Linear probing) | | | | |
| Pre-training Domain(DomainNet) | Painting∪Real∪Sketch | | | |
| Target Domain(OfficeHome) | Art | Clipart | Product | *avg.* |
| MAE | 29.00 | 36.65 | 43.12 | 36.26 |
| MAE+SWAD | 51.00 | 35.86 | 65.34 | 50.73 |
| DisMAE | 40.00 | 58.76 | 53.17 | 50.64 |
| DisMAE+SWAD | 40.00 | 60.36 | 44.18 | 48.18 |
| BSS | 40.00 | 55.58 | 53.97 | 49.85 |
| BSS+SWAD | 41.5 | 55.38 | 51.59 | 49.49 |
| MS-UDG | 50.00 | 67.33 | 63.76 | 63.76 |
| MS-UDG+SWAD | 52.00 | 67.73 | 65.34 | 61.69 |

