# OpenReview forum: "Minimal Semantic Sufficiency Meets Unsupervised Domain Generalization"
_NeurIPS.cc/2025/Conference — NeurIPS 2025 poster_

### Official Review · Reviewer_Ckt1 · 2025-06-27

**Clarity:** 2
**Significance:** 2
**Originality:** 2
**Rating:** 4
**Confidence:** 2

**Summary:**

This manuscript introduces MS-UDG, an unsupervised domain generalization method that learns minimal semantic representations by disentangling semantics from variations without requiring labels. While it shows improved performance over prior SSL and UDG methods, its contributions are largely incremental, with limited novelty. Additionally, inaccuracies in table annotations and weak theoretical insights diminish its overall contribution.

**Questions:**

Please refer to the weakness for more questions. I am open to changing the score if the concerns are well addressed.

Additional question

1. The concepts of sufficiency and minimality are directly adopted from existing work. It remains unclear what novel theoretical or practical contributions this manuscript provides beyond these established definitions.

**Ethical Concerns:**

["NO or VERY MINOR ethics concerns only"]

**Final Justification:**

Thank the authors for their response. While I appreciate the clarifications provided, using an existing theoretical solution in SSL for the first time in DG does not, in my view, constitute a strong theoretical/practical contribution, especially since SSL has already been justified in tackling DG problems in previous works. The contribution does not yet meet the bar expected for NeurIPS on my side, and I do not believe that this work will have a noticeable impact to the community.  Nonetheless, after carefully considering the completeness of this work and the ratings from other reviewers, I can increase my score to borderline acceptance.

**Limitations:**

yes

**Quality:**

2

**Strengths And Weaknesses:**

Strength
1. The experimental evaluation is comprehensive, including both quantitative benchmarks and qualitative visualizations.
2. The paper attempts to provide theoretical foundations for the proposed contrastive learning framework.

Weakness
1. The manuscript offers limited theoretical novelty, as it relies on the well-established concept of Minimal Sufficient Semantic Representation (Line-145) without introducing new insights. Moreover, information-theoretic frameworks have already been extensively explored in domain generalization [#1,#2,#3,#4]. The proposed method exhibits little distinction from prior work, as it merely applies existing ideas without providing any theoretical advancements.
2. Compared to existing work [#5], this manuscript primarily proposes incremental improvements, including introducing a mask, applying self-attention, and incorporating additional features for contrastive learning. Overall, the manuscript reads more like an extension or combination of [#5] and [#6] rather than presenting fundamentally new contributions.
3. In Table 1, under the Label Fraction: 10% setting, MS-UDG achieves the second-best performance in the photo and art domains, yet the authors mark it as the best. Similar inconsistencies appear in other parts of the manuscript, where MS-UDG results are incorrectly highlighted as the top-performing method despite ranking second. These misannotations may mislead readers and should be carefully corrected.
4. In the ablation study shown in Table 5 for the sketch domain, the performance when all losses are combined is worse than using each loss individually.
5. In Figure 3, the t-SNE visualization shows that the semantic features from MS-UDG exhibit poor category separability.
6. It is well known that self-supervised learning (SSL) benefits unsupervised tasks; this is not a novel contribution to the manuscript. The claim of operating without domain and category labels is also expected in the unsupervised setting and should not be regarded as a unique contribution.

[#1] INSURE: an Information theory iNspired diSentanglement and pURification modEl for domain generalization[J]. IEEE Transactions on Image Processing, 2024.

[#2] Rethinking domain generalization: Discriminability and generalizability[J]. IEEE Transactions on Circuits and Systems for Video Technology, 2024.

[#3] How Does Distribution Matching Help Domain Generalization: An Information-theoretic Analysis[J]. IEEE Transactions on Information Theory, 2025.

[#4] Invariant information bottleneck for domain generalization[C]//Proceedings of the AAAI Conference on Artificial Intelligence. 2022

[#5] Towards unsupervised domain generalization[C]//Proceedings of the IEEE/CVF Conference on Computer Vision and Pattern Recognition. 2022

[#6] Rethinking minimal sufficient representation in contrastive learning. In Proceedings of the IEEE/CVF Conference on Computer Vision and Pattern Recognition, pages 16041–16050, 2022

---

> ### Author Rebuttal · Authors · 2025-07-31
>
> We thank the reviewer for the detailed and insightful feedback on our manuscript. We reply on each point raised and address them as follows:
>
> **W1,W6,Q1: Theoretical novelty**
>
> **W1,W6,Q1-Ans.** First, we clarify that minimal sufficient representation builds on the established statistical concept of minimal sufficient statistics. Information bottleneck (IB) previously applied these principles to supervised learning, addressing the research problems of that time. Some methods extend this to the SSL scenario to discuss learning a minimal sufficient representation in unsupervised learning.
>
> Our framework adapts minimal sufficient statistics for learning domain-invariant representations in unsupervised domain generalization (UDG).  Crucially, UDG and supervised DG face fundamentally different theoretical challenges. PAC learning risk bounds for supervised domain generalization don't apply to self-supervised models where labels aren't available during training. This theoretical gap requires new frameworks tailored to the unsupervised setting. **Our work provides the first information-theoretic approach developed specifically for UDG.** Two learning objectives are induced from our theoretical framework, with empirical results validating this theoretical foundation across multiple benchmarks. In addition, **our framework first relates the representation in SSL model to the downstream generalization bounds in UDG.**
>
> In the end, we appreciate the reviewer thinking about the theoretical improvement. Our theoretical discovery on UDG parallels how IB-based methods [7,8] use the same statistical foundation for domain generalization—these works appeared at top conferences and demonstrated how classical concepts can tackle modern problems. In addition, the analogy can extend to methodological relationships: ERM(empirical risk minimization,i.e., CrossEntropy for classification) provides the baseline for domain generalization methods, while SSL serves as the foundation for UDG approaches like ours. We will clarify and emphasize the unique contributions in the revised version.
>
> **W2: Technical improvements**
>
> **W2-Ans.** We thank the reviewer for the detailed technical discussion. Our paper's core contributions span two interconnected areas: 1) information-theoretically grounded optimization objectives for UDG; 2) practical algorithms that implement these theoretical insights.
>
> Our approach differs fundamentally from existing UDG methods. The UDG baseline [5] employs adversarial training with domain labels to achieve domain invariance. This strategy operates independently of information-theoretic principles and targets different learning objectives than our framework. Meanwhile, H. Wang et al. [6] focus on capturing non-shared task-relevant information across domains to avoid overfitting, which contrasts with our goal of disentangling semantic content within the shared representation space for domain generalization.
>
> These methodological differences reflect deeper theoretical distinctions. While previous work addresses domain invariance through adversarial objectives or captures domain-specific information, our framework tackles the problem by formalizing what constitutes an optimal representation for generalization: one that preserves semantic sufficiency while achieving minimality. This information-theoretic perspective leads to novel optimization strategies that haven't been explored in prior UDG, DG, and SSL literatures.
>
> **W3,W4: 1. Misannotation and 2. table explaination**
>
> **W3,W4-Ans.**
>
> Misannotation: We thank the reviewer for identifying the annotation errors. These have been corrected in the revised manuscript.
>
> Table explaination: The tabulated results demonstrate our method's effectiveness across domains. Our loss design aims to improve generalization consistently across diverse domains rather than optimizing for individual domain performance. While the sketch domain shows a marginal decrease when using the complete loss formulation versus $\mathcal{L}_{max}$ alone, the remaining domains (photo, art, cartoon) all demonstrate substantial improvements. This trade-off yields the highest overall average accuracy (45.26), confirming that our unified framework successfully balances performance across the full domain spectrum rather than overfitting to specific domain characteristics.
>
> **W5: Explanation of the t-SNE visualization**
>
> **W5-Ans.** The t-SNE plots examine domain invariance properties by showing how our semantic representations $\mathcal{S}$ exhibit reduced domain clustering compared to baseline methods. The representation $v$ and vanilla MAE representations display clear domain separability, which validates our disentanglement approach, domain-specific information concentrates in the variation subspace, while semantic content becomes more domain-agnostic. This analysis targets the fundamental question of whether our method successfully separates domain-related variations from semantic content, rather than evaluating downstream classification performance. The visualization demonstrates that semantic representations avoid the domain-specific clustering patterns that typically hinder generalization, which aligns with our theoretical framework's emphasis on learning domain-invariant semantic features.
>
>
> [1] INSURE: an Information theory iNspired diSentanglement and pURification modEl for domain generalization[J]. IEEE Transactions on Image Processing, 2024.
>
> [2] Rethinking domain generalization: Discriminability and generalizability[J]. IEEE Transactions on Circuits and Systems for Video Technology, 2024.
>
> [3] How Does Distribution Matching Help Domain Generalization: An Information-theoretic Analysis[J]. IEEE Transactions on Information Theory, 2025.
>
> [4] Invariant information bottleneck for domain generalization[C]//Proceedings of the AAAI Conference on Artificial Intelligence. 2022
>
> [5] Towards unsupervised domain generalization[C]//Proceedings of the IEEE/CVF Conference on Computer Vision and Pattern Recognition. 2022
>
> [6] Rethinking minimal sufficient representation in contrastive learning. In Proceedings of the IEEE/CVF Conference on Computer Vision and Pattern Recognition, pages 16041–16050, 2022
>
>
> [7] Ahuja, Kartik, et al. "Invariance principle meets information bottleneck for out-of-distribution generalization." Advances in Neural Information Processing Systems 34 (2021): 3438-3450.
>
> [8] Du, Yingjun, et al. "Learning to learn with variational information bottleneck for domain generalization." European conference on computer vision. Cham: Springer International Publishing, 2020.

---

> > ### Comment · Reviewer_Ckt1 · 2025-08-05
> >
> > Thank the authors for their response. While I appreciate the clarifications provided, using an existing theoretical solution in SSL for the first time in DG does not, in my view, constitute a strong theoretical/practical contribution, especially since SSL has already been justified in tackling DG problems in previous works. The contribution does not yet meet the bar expected for NeurIPS on my side, and I do not believe that this work will have a noticeable impact to the community.
> >
> > Nonetheless, after carefully considering the completeness of this work and the ratings from other reviewers, I can increase my score to borderline acceptance. It is suggested to highlight the potential impact of this work.

---

> > > ### Author Response · Authors · 2025-08-05
> > >
> > > We appreciate the reviewer for the thoughtful suggestion. We will highlight the potential impact of our work in the revised manuscript.

---

### Official Review · Reviewer_yW3M · 2025-06-30

**Clarity:** 3
**Significance:** 2
**Originality:** 3
**Rating:** 4
**Confidence:** 2

**Summary:**

Authors propose an Unsupervised Domain Generalization (UDG) algorithm with theoretical analyses from the perspective of information theory. They also conduct extensive experiments to show its effectiveness.

**Questions:**

Please refer to weaknesses.

**Ethical Concerns:**

["NO or VERY MINOR ethics concerns only"]

**Final Justification:**

My concerns are well addressed in rebuttal, so I decide to raise my score.

**Limitations:**

Yes

**Quality:**

3

**Strengths And Weaknesses:**

Strengths:

- There are theoretical justifications for the proposed algorithm.
- Experimental results clearly demonstrate the effectiveness of the proposed algorithm.

Weaknesses:

- Newer and more recent SSL methods could be added as baselines, e.g. MoCo-v3 [1], SMoG [2], ReLICv2 [3], BAM [4]



[1] Chen X, Xie S, He K. An empirical study of training self-supervised vision transformers[C]//Proceedings of the IEEE/CVF international conference on computer vision. 2021: 9640-9649.

[2] Pang B, Zhang Y, Li Y, et al. Unsupervised visual representation learning by synchronous momentum grouping[C]//European Conference on Computer Vision. Cham: Springer Nature Switzerland, 2022: 265-282.

[3] Tomasev N, Bica I, McWilliams B, et al. Pushing the limits of self-supervised ResNets: Can we outperform supervised learning without labels on ImageNet?[J]. arXiv preprint arXiv:2201.05119, 2022.

[4] Shalam D, Korman S. Unsupervised Representation Learning by Balanced Self Attention Matching[C]//European Conference on Computer Vision. Cham: Springer Nature Switzerland, 2024: 269-285.

---

> ### Author Rebuttal · Authors · 2025-07-31
>
> We thank the reviewer for the positive recognition of the theoretical justification and the effectiveness of our proposed algorithm based on the experimental results.
>
> Regarding the suggestion to include newer SSL methods such as MoCo-v3 [1], SMoG [2], ReLICv2 [3], and BAM [4] as additional baselines, we appreciate this valuable recommendation. We have conducted additional experiments incorporating SSL methods MoCo-v3, SMoG, and BAM within UDG settings, as outlined below. Our method demonstrates competitive performance when compared to these SSL approaches.
>
> Regarding ReLICv2, the publicly available implementation provides only linear evaluation components without the complete SSL pretraining pipeline. Despite our efforts to contact the original authors for implementation details, time constraints prevented us from developing a full reimplementation from scratch. We acknowledge this limitation in our current evaluation and commit to including ReLICv2 in future benchmark extensions once the implementation becomes available.
>
> We plan to publicly release our UDG benchmark implementations for these SSL methods to facilitate future research and ensure reproducible comparisons within the community.
>
> All reported experiments utilize the official codebases of the respective SSL methods, adapted to conform with standard UDG evaluation protocols while maintaining their core algorithmic components.
>
> | target domain | photo | art   | cartoon | sketch | avg   |
> |---------------|-------|-------|---------|--------|-------|
> | SMoG          | 70.81 | 48.72 | 45.76   | 48.71  | 53.50 |
> | BAM           | 52.16 | 54.33 | 46.72   | 58.89  | 53.03 |
> | MoCo-v3        | 70.61 | 64.10 | 54.03   | 52.21  | 60.24 |
> | Ours          | 74.10 | 61.90  | 63.73   | 73.66  | 68.35 |
>
> Table 1. The results on PACS under 10% label fraction. We report the accuracy for every domain and the average accuracy for all domains.
>
> | target domain | clipart | infograph | quickdraw | painting | real  | sketch | overall | avg.  |
> |---------------|---------|-----------|-----------|----------|-------|--------|---------|-------|
> | SMoG          | 41.75   | 17.41     | 17.15     | 24.22    | 35.50 | 26.52  | 26.37   | 27.09 |
> | BAM           | 74.96   | 30.88     | 25.66     | 37.07    | 46.24 | 47.67  | 43.98   | 43.75 |
> | MoCo-v3        | 74.90   | 30.56     | 30.98     | 52.54    | 62.31 | 52.85  | 49.33   | 50.69 |
> | Ours          | 79.70   | 30.01     | 40.11     | 53.73    | 63.77 | 65.82  | 53.37   | 55.52 |
>
> Table 2. The results on DomainNet under 10% label fraction. We report the accuracy for every domain and the average accuracy for all domains.

---

> > ### Comment · Reviewer_yW3M · 2025-08-06
> >
> > Thanks for your rebuttal. I decide to raise my score to 4.

---

> > > ### Author Response · Authors · 2025-08-07
> > >
> > > Thank you very much for the feedback.  We truly appreciate your time and valuable input.

---

### Official Review · Reviewer_sD9g · 2025-07-02

**Clarity:** 3
**Significance:** 3
**Originality:** 4
**Rating:** 5
**Confidence:** 3

**Summary:**

This paper explores Unsupervised Domain Generalization (UDG), a task aimed at improving the generalization of unsupervised learning models like Self-Supervised Learning (SSL) without using category labels. While prior UDG methods rely on domain labels to separate semantics from variations, this approach is limited by the unavailability of such labels in real-world scenarios. The authors propose a new framework called Minimal-Sufficient UDG (MS-UDG), which learns semantic representations by optimizing for both sufficiency (retaining shared semantic information) and minimality (removing irrelevant variations). Based on information theory, MS-UDG combines an InfoNCE-based objective with novel components to disentangle semantics and variations, achieving state-of-the-art performance on UDG benchmarks without needing category or domain labels.

**Questions:**

Please refer to the Weaknesses.

**Ethical Concerns:**

["NO or VERY MINOR ethics concerns only"]

**Final Justification:**

My issues are addressed. After reading all reviewers' comments and authors responses,  I prefer to maintain my initial rating.

**Limitations:**

yes

**Quality:**

3

**Strengths And Weaknesses:**

Strengths:

1. This paper is well motivated.

2. The experimental results are good.

3. Overall, the proposed method is simple and effetive.

Weaknesses:

1. Line 166 I(x_1;x_2;T) = I(x_1;x_2;S), what does S stands for? According to the following text, S may stand for semantic. But what does semantic information in X_{ssl} means? SSL should contain no label, so it hard to understand.

2. Line 116~117 downstream supervised data P(X_{sup}, Y_{sup}) is used for fine-tuning? Is it the source domain ine Line 270?

---

> ### Author Rebuttal · Authors · 2025-07-31
>
> We thank the reviewer for the thoughtful and positive feedback, including sound motivation, effectiveness of the method ,and promising experimental results. Below, we address each comment and question in detail.
>
> **W1: Explanation of semantic factor $\mathcal{S}$**
>
> **W1-Ans.** While self-supervised models cannot anticipate specific downstream tasks during pretraining, we can systematically identify categories of information that consistently impede cross-domain generalization, such as task-irrelevant information. In UDG settings, task-irrelevant information typically encompasses domain-specific characteristics such as imaging style, background patterns, lighting conditions, and acquisition artifacts, collectively represented as $\mathcal{V}$ in our framework. Given our decomposition $\mathcal{Z}=\mathcal{S}\oplus\mathcal{V}$, the semantic component $\mathcal{S}$ captures the complementary information: fundamental visual structures, object relationships, and content-based features that remain consistent across domains and prove valuable for downstream recognition tasks.
>
> This principled separation between semantic content and domain variations builds upon established theoretical foundations in UDG literature [1,2]. The practical significance becomes evident in real-world scenarios where training data inevitably spans multiple domains. For instance, medical datasets may aggregate scans from different institutions, each using varied imaging protocols, scanner manufacturers, and acquisition parameters. The resulting domain shifts introduce spurious correlations that can mislead models away from clinically relevant features. Our framework addresses these challenges by explicitly optimizing representations to retain semantic sufficiency while systematically reducing dependence on domain-specific information.
>
> **W2: Clarify the finetuning data**
>
> **W2-Ans.** Both the SSL pretraining data $X_{ssl}$ and supervised finetuning data $X_{sup}$ originate from source domains, following the standard UDG protocol where no target domain data is accessible during training. This setup mirrors classical domain generalization but operates without category labels during the representation learning phase. The key challenge lies in learning generalizable features from source domain variations that transfer effectively to unseen target distributions at test time. Our experimental setup strictly adheres to the benchmark protocols established in [1,2], ensuring fair comparison with existing methods. Specifically, we use the same train/validation/test splits and domain partitions as prior UDG works, where source domains provide both unlabeled data for SSL pretraining and labeled data for downstream task finetuning, while target domains remain completely held out until evaluation.
>
>
> [1] Zhang, An, et al. "Disentangling Masked Autoencoders for Unsupervised Domain Generalization." European Conference on Computer Vision. Cham: Springer Nature Switzerland, 2024.
>
> [2] Zhang, Xingxuan, et al. "Towards unsupervised domain generalization." Proceedings of the IEEE/CVF Conference on Computer Vision and Pattern Recognition. 2022.

---

### Official Review · Reviewer_Uubv · 2025-07-02

**Clarity:** 4
**Significance:** 4
**Originality:** 4
**Rating:** 5
**Confidence:** 1

**Summary:**

The paper addresses Unsupervised Domain Generalization (UDG) for image classification by formalizing the task through an information-theoretic framework. The authors propose learning Minimal Sufficient Semantic Representations that capture shared semantic information across multiple augmented views of the same image while eliminating semantically irrelevant information. The approach includes theoretical proof demonstrating that this formulation reduces out-of-distribution risk, which directly addresses the UDG challenge.

**Questions:**

Can you provide failure case analysis explaining why certain competing methods outperform MS-UDG in specific scenarios, and what insights does this offer about your method's limitations ?

**Ethical Concerns:**

["NO or VERY MINOR ethics concerns only"]

**Final Justification:**

I appreciate the authors' response addressing my concerns regar sectionding the limitations of MS-UDG in some specific scenarios. The provided explanation has resolved my initial question, and I will therefore keep my original score.

**Limitations:**

yes.

**Paper Formatting Concerns:**

no concerns.

**Quality:**

4

**Strengths And Weaknesses:**

**Strengths:**

- **Writing Quality** - The paper is exceptionally well-written with clear and well-motivated technical components.
- **Comprehensive Evaluation** - The experimental comparison against existing methods is extensive and thorough, providing strong empirical validation.
- **Complete Ablation** - The ablation studies effectively justify the use of both loss components, demonstrating that their combination yields superior performance.
- **Theoretical Foundation** - The paper provides comprehensive theoretical analysis, introducing all necessary concepts regarding minimal sufficient semantic representations before presenting the method and its implementation.

**Weaknesses:**

- **Missing Failure Analysis** - The paper lacks a failure analysis section examining cases where competing methods outperform MS-UDG. Such analysis would provide valuable insights into the method's limitations and help readers understand when and why the approach may not be optimal.

---

> ### Author Rebuttal · Authors · 2025-07-31
>
> We appreciate the reviewer for the positive recognition of our work, including the writing quality, comprehensive evaluation, complete ablation studies, and theoretical foundation. Below, we provide a detailed analysis of the failure cases and discuss the insights gained from competing methods that sometimes outperform MS-UDG in specific scenarios.
>
> **Missing Failure Analysis**
> We conducted a comparative analysis with two advanced UDG methods, BSS and CycleMAE. The results highlight the strengths and limitations of each approach.
>
> * BSS (Fourier Augmentation Method):
> BSS benefits from mixing domain information through augmentation techniques, which sometimes enables it to outperform our method in domains with simpler and more uniform textures, such as sketches, quickdraw, and clipart. This improvement may not be suitable for rich-style domains, like art and real, since these domains contain richer low-frequency information compared with other domains. The Fourier augmentation in BSS may not significantly alter the style, which limits the effectiveness of data augmentation. Without disentanglement or other generalization techniques, the performance of BSS on rich-style domains is often inferior to that on other domains.
>
> * CycleMAE (Generation-Based Method):
> Generation-based methods, such as CycleMAE, learn informative representations through reconstruction. However, domains like sketches often result in trivially low values of reconstruction losses because of simple backgrounds and domain styles. As a result, CycleMAE tends to focus on learning representations from richer-style domains, such as real images, art, and infographics. This bias toward more complex domains can cause CycleMAE to outperform MS-UDG in those specific areas, while in simpler-style domains, it underperforms compared to MS-UDG..
>
> Even though these two methods occasionally perform better than our method through their own advantages, our MS-UDG method strikes a balance between these two approaches, which can enhance the generalization ability.  $L_{max}$ aims to learn the domain-related representation through reconstruction and $L_{min}$ helps disentangle the domain-related information from sufficient semantic representations.   As a result, our method achieves state-of-the-art performance on average, while still maintaining robustness across a diverse set of domains.
>
> We will add these discussions to the revised manuscript. Thank you for the advice for better insights into those methods.

---

> > ### Comment · Reviewer_Uubv · 2025-08-06
> >
> > I appreciate the authors' response addressing my concerns regar sectionding the limitations of MS-UDG in some specific scenarios. The provided explanation has resolved my initial question.

---

> > > ### Author Response · Authors · 2025-08-07
> > >
> > > Thank you for your thoughtful feedback. We sincerely appreciate your time and insightful comments on our work.

---

### Decision · Program_Chairs · 2025-09-17

**Decision:**

Accept (poster)

**Comment:**

The paper introduces a new method for unsupervised domain generalization (UDG), building upon contrastive learning principles but proposing explicit loss components to achieve minimal sufficient semantic disentanglement. The authors presents their method from an information-theoretic perspective, distinguishing between semantic and variation components of the learned representation, and to optimize explicitly for minimal sufficient semantics. All reviewers agree that this is a solid paper with thorough experimental results and sufficiently well-motivated theory. The empirical evaluation is comprehensive, and the theoretical grounding connects well with the proposed training objectives. The ablation studies also demonstrate the usefulness of the loss components. The reviewer comments were also well addressed during the rebuttal. Overall, this is a good contribution for the important problem of semantic disentanglement and unsupervised domain generalization, a nice balance between reasonably-well motivated theory and training loss components that seem to demonstrate good experimental results.